# The Study of the Caudal Vertebrae of Thick-Toed Geckos after a Prolonged Space Flight by X-ray Phase-Contrast Micro-CT

**DOI:** 10.3390/cells12192415

**Published:** 2023-10-07

**Authors:** Inna Bukreeva, Victoria I. Gulimova, Yuri S. Krivonosov, Alexey V. Buzmakov, Olga Junemann, Alessia Cedola, Michela Fratini, Laura Maugeri, Ginevra Begani Provinciali, Francesca Palermo, Alessia Sanna, Nicola Pieroni, Victor E. Asadchikov, Sergey V. Saveliev

**Affiliations:** 1Institute of Nanotechnology, CNR, Rome Unit, Piazzale Aldo Moro 5, 00185 Rome, Italy; inna.bukreeva@cnr.it (I.B.); ojunemann@yandex.ru (O.J.); alessia.cedola@cnr.it (A.C.); michela.fratini@gmail.com (M.F.);; 2P.N. Lebedev Physical Institute Russian Academy of Sciences, Leninskiy Prospekt 53, 119991 Moscow, Russia; 3Avtsyn Research Institute of Human Morphology of Federal State Budgetary Scientific Institution, “Petrovsky National Research Centre of Surgery”, Tsyurupy Str. 3, 117418 Moscow, Russia; braincase@yandex.ru; 4Federal Scientific Research Centre “Crystallography and Photonics”, Russian Academy of Sciences, Leninskiy Prospekt 59, 119333 Moscow, Russiaasad@ns.crys.ras.ru (V.E.A.); 5IRCCS Fondazione Santa Lucia, Via Ardeatina 306/354, 00142 Roma, Italy; 6Physics Department, ‘Sapienza’ University, Piazzale Aldo Moro 2, 00185 Rome, Italy; 7Laboratoire d’Optique Appliquée, CNRS, ENSTA Paris, Ecole Polytechnique IP Paris, 91120 Palaiseau, France

**Keywords:** spaceflight adaptation, Bion-M1, thick-toed gecko (*Chondrodactylus turneri*, Gray, 1864), proximal caudal vertebrae, notochord, intercentrae, X-ray phase-contrast tomography, microtomography

## Abstract

The proximal caudal vertebrae and notochord in thick-toed geckos (TG) (*Chondrodactylus turneri*, Gray, 1864) were investigated after a 30-day space flight onboard the biosatellite Bion-M1. This region has not been explored in previous studies. Our research focused on finding sites most affected by demineralization caused by microgravity (G0). We used X-ray phase-contrast tomography to study TG samples without invasive prior preparation to clarify our previous findings on the resistance of TG’s bones to demineralization in G0. The results of the present study confirmed that geckos are capable of preserving bone mass after flight, as neither cortical nor trabecular bone volume fraction showed statistically significant changes after flight. On the other hand, we observed a clear decrease in the mineralization of the notochordal septum and a substantial rise in intercentrum volume following the flight. To monitor TG’s mineral metabolism in G0, we propose to measure the volume of mineralized tissue in the notochordal septum. This technique holds promise as a sensitive approach to track the demineralization process in G0, given that the volume of calcification within the septum is limited, making it easy to detect even slight changes in mineral content.

## 1. Introduction

In our research, the proximal caudal vertebrae and notochord of thick-toed geckos (TG) (*Chondrodactylus turneri*, Gray, 1864) were studied following a 30-day space flight on an unmanned spacecraft (USC) “BION-M1” (19 April–19 May 2013). Flotation is a significant concern in space flights, causing humans and animals to lose support afferentation and to experience decreased mechanical loading, hypokinesia, and radiation exposure, which can lead to microgravity (G0)-induced muscle and bone atrophy during long-duration space flights [1,2,3,4,5,6]. Geckos are a unique natural model able to maintain nearly normal behavior and locomotion even in weightlessness [7,8,9]. Their ability to climb vertical surfaces and attach to objects [10,11] helps them maintain normal activities and behaviors aboard spacecraft that keep their bones and muscles conditioned [12,13,14,15].

The present study, based on X-ray phase-contrast micro-tomography (XPCT), examines the changes caused by G0 in the proximal caudal part of the vertebral column of geckos, particularly in their trabecular and cortical compartments and notochords. In our research, we found no significant G0-induced bone deficits in the vertebrae, which indicates good adaptability of geckos to G0 conditions during the 30-day space flight. The study of gecko’s chondrocyte mineralization in the notochord provides a novel sensitive method for examining the effect of G0 on biomineralization. Our research indicates that prolonged flight potentially could reduce the mineralization of notochordal cartilage of geckos. These important scientific results must be examined in more detail in future space flight missions with geckos on board.

Microgravity has profound effects on the physiology of living organisms, particularly on the health of bones [16,17,18]. Bones are complex tissues that remodel continuously even in normal physiological conditions [19]. Researchers have found that G0 is associated with a significant reduction in bone density [20,21,22]. In fact, astronauts can lose up to 1–2% of their bone mass per month during extended periods of space flight [23]. Bone loss is thought to result from skeletal unloading in G0 that disrupts the resorption–formation balance, causing a shift in the balance towards resorption [2,6]. Therefore, the impact of G0 on bone mineral metabolism due to the mechanical unloading of bone is critical to addressing the serious challenges astronauts face during and after space flight [2,16,17,23].

Animal models have played a crucial role in space research over the years [24,25,26,27,28,29]. Among them, rodents remain the most commonly used model objects [30,31,32,33]. According to scientific research, mature mice that were exposed to G0 have undergone substantial bone loss, which was particularly evident in load-bearing sites [34]. The bone analysis of mature male C57/BL6 mice after a 30-day Bion-M1 mission showed that long-term space flight impacts the bone microstructure, tissue-level mechanical properties, osteocyte survival, and lacunae volume [21]. An analysis of the mice’s spine via micro-CT showed that long-term space flight affects the vertebral column in a site-specific manner, possibly due to the different load on different parts of the spine [21]. The lumbar and caudal vertebral bodies displayed a loss of bone [21,24]. On the other hand, the thoracic vertebral body remained unaffected, presumably due to mechanical respiratory movements.

Geckos have proven to be highly adaptable to space flight conditions, as demonstrated in 16- and 12-day orbital experiments on unmanned spacecraft (USC) series Foton-M2 and Foton-M3 and after a 30-day space flight on USC Bion-M1 [7,8,12,13]. They have the ability to survive for long periods while using minimal resources, which makes them a valuable model for space exploration. In addition, geckos’ exceptional ability to climb vertical surfaces and attach to objects enables them to exhibit nearly normal behavior and locomotion even in G0 environments. Two different research teams, using four types of CT, came up with results, indicating that the gecko’s attachment ability could have a positive impact on mineral metabolism and potentially reduce demineralization in G0 [8]. This fact has considerable scientific value in the study of G0-induced bone loss.

Geckos have a skeletal bone organization similar to that of other vertebrates. Reptiles belong to amniotes; thus, their bones have a closer mineral metabolism to mammals than lower vertebrates. Throughout life, geckos maintain their notochord (a stiffening rod that serves as the primary axial skeleton during embryonic development). A notochord in geckos is formed of vacuolated cells and is chondrified at mid-vertebral locations to form the notochordal cartilage or septum. The adult notochord of a gecko can eventually mineralize [35,36].

We used XPCT imaging in free-space propagation mode to study the 3D internal structure of proximal caudal vertebrae of geckos. This advanced imaging method allowed us to visualize the whole vertebrae down to the cellular level without having to perform any destructive sample preparations, such as bone and soft tissue dissection, bone drying, or tissue staining. Conventional CT scanning is not suitable for imaging very small structures, such as single cells, due to the poor absorption of soft tissue. On the other hand, XPCT can provide information about the phase shift caused by objects, which allows for the improvement of image contrast and enables the high-resolution 3D imaging of both bone and soft tissue [37,38].

Our studies of proximal caudal vertebrae of geckos after 30-day flights showed no significant signs of demineralization that are common for other animals and humans during prolonged periods of weightlessness. In particular, we did not detect significant changes in the basic morphometric parameters, such as bone volume fraction (BV/TV, *p* > 0.3), cortical bone volume fraction (Ct.BV/TV, *p* > 0.2), and trabecular bone volume fraction (Tb.BV/Sc.V, *p* > 0.7). These findings were supported by the Foton-M series and Bion-M1 bone experiments.

On the other hand, we observed changes in the mineralization of notochordal cartilage (septum) and intercentrum. In particular, we found a statistically significant decrease in the calcified tissue fraction of the septa (−77.83%, *p* < 0.001) after space flight. At the same time, a significant increase in bone volume (BV) in the intercentra was found by +35.73% (*p* = 0.0343) in the flight group vs. the control.

The approach we suggest in this study involves examining the notochordal septa calcification to observe the impact of G0 on mineral metabolism. We believe that the degree of septum mineralization may be a sensitive indicator of the mineral balance response to G0 exposure compared to bone loss. A small amount of calcified tissue in the septum makes it easier to detect even small changes in the mineral balance, whereas these changes can be unobvious in bones, given their high degree of mineralization. Our novel approach could prove to be beneficial in space exploration related to bone loss.

## 2. Material and Methods

### 2.1. Animals and Life-Support

This study focused on thick-toed geckos (*Chondrodactylus turneri* GRAY 1864) and utilized mature females aged 1.5–2 years. The average weight of the studied geckos was 20.0 ± 1.5 g, with an average snout-vent length of 78.9 ± 2.3 mm and an average total length of 154.3 ± 6.5 mm. To facilitate identification in both day and night conditions, the geckos were marked with colored collars.

Each gecko group ultimately comprised 5 females. To ensure stable social relations and minimize aggressive behavior during this study, four preliminary groups of six females were formed for both the flight and ground delayed synchronous control (GDSC) experiments. In this way, any aggressive gecko or entire group could be excluded without the need for individual replacements.

A total of 15 geckos were taken on an orbital 30-day space flight (F) aboard the BION-M No. 1 biosatellite (19 April–19 May 2013). The geckos were housed in three research and support blocks (RSBs), each holding five geckos. The RSBs had a volume of 5.9 L each. Five tubular gecko shelters made of American oak were mounted on the walls lined with hardboard (a type of fiberboard). The floor was covered with textile laminate (fabric-reinforced laminate). RSBs had a hole in the floor for a revolving-type feedbox that was plugged when not in use. The feedbox was equipped with ten cavities for geckos’ food. The geckos were fed a mixture of mealworms (*Tenebrio molitor*) with bran, dried carrots, crushed eggshells, and potable gel particles. The feedbox was opened every third day, beginning on the launch day, for four hours. On the floor, there were two 5 cm diameter heating zones. During the daytime, these zones kept a local surface temperature of 31 to 32 °C. There were 8.00–20.00 daylight hours and 20.00–24.00 nighttime hours. The temperature on board averaged 21 to 22 °C. LEDs, a video camera, and a fan were mounted on the RSB lid. The bottom of the RSB was illuminated by approximately 485 lx during the daytime and 8 lx at night. The ventilation fan flowed continuously at about 3.0 L per minute.

The GDSC experiment was carried out with 15 female thick-toed geckos on the ground in similar RSBs, keeping the same temperature and CO_2_/O_2_ levels as the space flight conditions. The GDSC experiment was scheduled 98 days after the space flight experiment and took place at the Institute of Biomedical Problems of the Russian Academy of Sciences, Moscow, Russia, from 27 July to 26 August 2013.

### 2.2. Logistics and Sample Preparation

Following the landing of the satellite in Orenburg (19 May 2013, 07/12 Moscow time), the containers with animals were unsealed, and all geckos were thoroughly inspected on the landing site. The physical state and appearance of the geckos had no visible alteration, and their capacity to attach to vertical and inverted surfaces was preserved. Within the next 13 h, the reptiles were transported to Moscow in marked individual plastic containers. Upon arrival, the geckos were euthanized by the intraperitoneal injection of Nembutal (4 mg per 20 g of body weight); then, their length and weight were measured. Subsequently, they were subjected to autopsies and preserved in Richard-Allan Scientific 10% neutral Buffered Formalin, pH 7.4. The fixed geckos were stored in this solution at +4–8 °C until 20 August 2018 (5 years and 3 months), when, as part of a macrodissection, fragments of the sacral vertebrae (0.5–2 sacral vertebrae) and the caudal vertebrae (2–5 caudal vertebrae) were isolated from the proximal tail. The more-distal sections of the tail were utilized in other experiments but were not conserved for future analysis.

On 22 August 2018, all samples were washed with distilled water for 1 h before being placed in marked containers with 70% alcohol. The prepared samples were transported and studied at the ID17 beamline of ESRF (Grenoble, France). The caudal vertebrae and intercentra were chosen as the areas of interest.

### 2.3. Ethical Statement

This study was approved by the Biomedical Ethics Commission of Russian Federation State Research Center—Institute of Biomedical Problems, Russian Academy of Sciences/Physiology Section of the Russian Bioethics Committee of Russian Federation National Commission for UNESCO (minute № 319 from 4 April 2013) and was conducted in compliance with the European Convention for the Protection of Vertebrate Animals used for Experimental and Other Scientific Purposes (1986).

### 2.4. X-ray Phase-Contrast (XPCT) Experiment

Three-dimensional studies of geckos were carried out using propagation-based imaging (PBI) setup [39] for X-ray phase-contrast tomography (XPCT) [40]. The approach relies on both attenuation and phase shifts of the X-ray beam transmitted through the sample. The technique allows the visualization of high- and low-absorbing biological tissues at different scales, down to the cellular level [41,42].

Data for 3D micrometer-resolution images of the samples were collected on the biomedical beamline ID17 of ESRF. The measurement was performed with a pink beam [43] with an energy of around 60 keV. The sample-to-detector distance was 1.5 m. A PCO.edge 5.5 sCMOS camera from Kelheim, Germany, was utilized with an effective pixel size of 6.1 × 6.1 micron^2^. To perform 3D imaging, 4000 projections of the sample were taken in extended field-of-view (FOV) mode and with a pixel size of 3.0 × 3.0 micron^2^. The extended FOV mode is an acquisition procedure that enables nearly twice the effective horizontal extent of the detector’s FOV. The sample rotation axis is shifted to the left or right side of the FOV, and a dataset of projections is collected over a full 360° rotation angle, with a size equal to the detector’s FOV. Following the appropriate stitching of sinograms, the reconstruction process can be carried out in a common way. Exposure time was 35 ms per projection.

Tomographic experiments at the sub-micron level were carried out at ID17 beamline at ESRF.

At the ID17 beamline, the experiment employed a filtered polychromatic beam with a peak energy of 60 keV and a sample-to-detector distance of 1 m. A PCO.edge 5.5 camera, with a 10 × Optique Peter from Lentilly, France, was utilized as a detector system. This camera achieved an effective pixel size of 0.7 × 0.7 micron^2^. The CT acquisition time was 50 ms per projection. For each region, we acquired 6000 projections.

### 2.5. Data Processing

Tomographic reconstruction was performed with the CGLS algebraic method using the ASTRA-TOOLBOX software package [44].

The open-source image processing software ImageJ/Fiji (version 1.54f) was employed for 3D visualization [45].

To assess the morphometric parameters of the vertebrae, primary segmentation of the bone tissue was performed based on X-ray phase-contrast tomography images using the machine learning software “Ilastik”(version 1.3.3post3) [46] under the supervision of biologists. More specifically, the “Ilastik” software was used in the “pixel classification” mode, in which the “Random Forests” [47] method is implemented as a classifier. Further segmentation of the cortical and trabecular bone tissue of the vertebrae was performed using morphological operations (dilation, erosion, closing, etc.) described in [48]. The segmentation steps of the vertebrae are illustrated in Appendix A.

### 2.6. List of Morphometric Parameters and Abbreviations

Figure 1A shows XPCT sections of the gecko’s vertebrae, with segmented cortical bone depicted in yellow and the subcortical area depicted in red.

The morphometric parameters, including total volume (TV is the volume of the entire region of interest), bone volume fraction (BV/TV), cortical bone volume fraction (Ct.BV/TV), and trabecular bone volume normalized by subcortical volume (Tb.BV/Sc.V, trabecular bone volume fraction) were calculated for each sample of the vertebra. Additionally, the septum mineralized cartilage volume (Sept.MCV) and septum mineralized cartilage volume fraction (Sept.MCV/TV) were calculated for the mineralized septa located in the notochordal canal of the vertebrae (Figure 1B). It should be noted that by subcortical volume (Sc.V), we mean the volume limited by the endocortical surface. To calculate the morphometric parameters, the entire cortical and trabecular bone compartments (Figure 1A) were selected in each ROI. The distance-transform algorithm described in [49] was used to calculate bone morphometric parameters, such as trabecular thickness (Tb.Th), trabecular separation (Tb.Sp), and trabecular number (Tb.N), in the ventral side of the vertebrae (see Figure 1A). The BV parameter was calculated for the intercentra.

The abbreviations used in this work as well as the methodology of calculation correspond to Refs [50,51]. All morphometric parameters of the samples were measured using Python (version 3.9). Parameters Tb.Th, Tb.Sp, and Tb.N were calculated with the Python Toolkit “Porespy”(version 2.3) [52].

### 2.7. Statistical Analysis

The differences in the mean value of tested parameters between the control and flight groups of TG are presented as a percentage, while the value of the parameters in the control group was taken as 100%. The Shapiro–Wilk Test was used to detect normally distributed values. Comparisons between the control and flight groups were performed using Welch’s *t*-test for normally distributed values and the Mann–Whitney U-test for non-normally distributed values. The significance level was set at *p* = 0.05 for all statistical tests. Statistical analysis was performed using the Python “scipy.stats” package (version 1.11) [53].

### 2.8. Histological Study

To assess the histological structure of analyzed samples, we used the most proximal fragment of the tail of the thick-toed gecko, taken directly rostral from the vent, 2 mm thick. The sample was removed and fixed in 10% neutral buffered formalin. Then, it was decalcified in neutral SoftiDec (BioVitrum, Moscow, Russia), dehydrated, and embedded in paraffin. After that, frontal 10 μm sections were prepared and stained with hematoxylin-eosin and Mallory’s trichrome stain.

## 3. Results

In this study, we investigated the effects of prolonged weightlessness on the spine and notochord in the caudal proximal region of the gecko’s vertebral column after a one-month space flight aboard the USC Bion-M1(Roscosmos, Moscow, Russia).

Thick-toed geckos (see Figure 1A), which are small reptiles, possess unique morphological characteristics, such as the particular structure of the skeleton and subdigital pads on their toes, which make them a valuable model for studying the effects of space flight on bone mineral balance and cellular activity.

Geckos’ ability to locomotion in complex habitats is largely due to the particular morphology of their spine. The concave amphicoelous vertebral structure with the persistent notochord and the presence of notochord-derived cartilage enhance flexibility for a greater range of motion. Spine flexibility, together with the adhesion ability of their pads to bind and climb on surfaces, helps geckos to maintain normal activities and behavior during weightlessness, avoiding the stress associated with floating.

Following the landing and euthanasia of the geckos, different sections of their bodies were allocated for the multiple intended scientific experiments. Three female adult geckos from the GDSC group and three from the flight group were used in our research for analyzing the proximal caudal part of the gecko’s spine, including the 2–5 th caudal vertebrae.

Figure 2a displays a photographic image of a female thick-toed gecko marked with a colored collar, which participated in the orbital experiment during the 30-day space flight of the Bion-M1(Roscosmos, Moscow, Russia) biosatellite. Figure 2b shows the posterior view of the thick-toed gecko’s caudal vertebra.

Histology was utilized to characterize the tissue and cellular composition of the TG caudal sample. Figure 3a shows a transverse section of TG tail, stained with Mallory’s stain, performed near the mid-vertebral region. The notochordal canal (nc); spinal cord (sc); and vertebra (v) are clearly visible in the figure.

In the gecko, a persistent notochord runs through the vertebral centra via the notochordal canal. Both the notochordal canal and the notochord display a moniliform morphology due to periodic constrictions along their length. This morphological feature is visible in Figure 3b, showing a sketch of a longitudinal section of the TG stained with Mallory stain. The notochord is composed primarily of alternating zones of chordoid (fibrous) and chondroid (cartilaginous) tissue. Chordoid tissue is observed throughout the entire span of the notochord and fills the cavities between amphicoelous vertebrae in the intervertebral joints. The chondroid tissue is, on the other hand, restricted to mid-vertebral regions corresponding to reduced diameters of the notochord.

Figure 3c,d shows the transverse section of the notochord with the chordoid tissue and chondroid tissue (Mallory stain). Chordoid tissue in Figure 3c mainly consists of large, irregular-shaped chordocytes with a center-located vacuole and sideways-located nucleus. Chordocytes are densely clustered with the scarce extracellular matrix. The chondroid tissue in Figure 3d contains many oval-shaped, non-vacuolated cells and a substantial amount of extracellular matrix.

### 3.1. XPCT Experimental Results and 3D Visualization

The XPCT technique was employed to produce a highly detailed 3D image of the TG’s body fragments, specifically focusing on the proximal caudal segment of the vertebral column. Before being scanned, the proximal caudal part was dissected from the body of TGs and measured at beamline ID17 of ESRF (Grenoble, France) without further invasive preparation. The tomographic data were used for the quantitative assessment of the morphometric parameters of the gecko’s spine and notochord.

The image shown in Figure 4a illustrates the tomographic slice of the entire TG sample (transverse section); the magnified view of the region boxed in Figure 4a is shown in Figure 4b (the vertebral and notochordal canals are indicated by thick and thin black arrows in the figure). XPCT imaging enables the observation of both calcified and soft tissues, as illustrated in Figure 3a,b, highlighting the skeletal structure, muscular system, and central nervous system.

Our research centered on the TG vertebral column and notochord of TG, with a particular emphasis on the alterations in their morphological structure induced by microgravity. The typical 3D view of the proximal caudal vertebrae studied in our research is shown in Figure 4c. The image was obtained via the virtual segmentation of bone in the XPCT image (see Section 2.5). Figure 4d displays the lateral cross-section of the vertebrae, comprising the notochordal canal (indicated by a thin white arrow) and the notochord containing mineralized notochordal cartilage–septum (indicated by a thick white arrow).

Mineralization is a natural process that leads to the accumulation of minerals, particularly calcium, in tissues. These processes can occur in the notochord, specifically in the chondroid tissue, of some gecko species. In the scientific literature, the calcified chondroid tissue in geckos is mainly associated with autotomic fracture planes, allowing for detachment and providing a scaffold for the regrowth of spinal tissues after autotomy.

Our research on thick-toed geckos has shown that the mineralization of notochordal cartilage (septum) is not limited to the autotomy plane of the caudal vertebrae but can also be found in the more anterior part of the spine, contributing to the overall structural integrity of the gecko’s skeleton.

To gain a deeper understanding of the mineralized notochordal cartilage structure, a tomographic scan with a pixel size of 0.7 × 0.7 μm^2^ was performed on the relevant parts of the specimen. Figure 5 shows the XPCT slice (lateral view (a) and transversal view (b)) of the vertebrae with the notochordal canal.

Our XPCT study, which is consistent with our histological results shown in Figure 3 and the results published in Ref. [35], suggests that the notochord is composed of chordoid tissue periodically separated by chondroid tissue with non-vacuolated oval-shaped cells and considerable amounts of extracellular matrix (see Figure 5). Chondroid tissue does not contain blood vessels, nerves, or lymphatic vessels. Additionally, it undergoes mineralization in the mid-vertebral areas, which corresponds to a reduction in notochord diameters (see the longitudinal section of a vertebra with notochordal canal and notochord in Figure 5a). The images in Figure 5b,c highlight the mineralized notochordal cartilage in a transversal section of the sample at the mid-vertebral regions. The white arrow in Figure 5b indicates chondrocytes, and red dots show chondrocyte nuclei. Figure 5c illustrates the zoomed image of the mineralized septum.

### 3.2. The Vertebrae and Notochord 3D Analysis

Our research involved analyzing the impact of G0 on the mineralization of both gecko bones and notochordal chondroid tissue. Our research suggests that the mineralization of the notochordal septum may be a sensitive indicator of mineral balance when compared to bone loss after G0 exposure. The septum is less mineralized; thus, even slight changes in mineral content can be detected,

A comparison between the control and flight groups was performed to estimate the impact of space flight on morphometric parameters of both gecko’s vertebrae and mineralized septa. The results of statistical data analysis are presented in the form of BoxPlot diagrams (see Figure 6) and in Table 1. There were no significant differences in the body mass of geckos between the control and flight groups either before flight (−3.72%, *p* > 0.6) or at landing (+2.9%, *p* > 0.6).

For the whole vertebrae, the total volume (TV) was calculated as a volume limited by the outer surface of the vertebrae. Statistically nonsignificant decreases in the TV parameter (−12.55%, *p* = 0.1608) were found after space flight. For the main bone parameters, such as bone volume fraction (BV/TV), cortical bone volume fraction (Ct.BV/TV), and trabecular bone volume fraction (Tb.BV/Sc.V), no significant differences (*p* > 0.2) were found between the control and flight groups.

After the experiment, no significant differences were found for the parameters of trabeculae, including trabecular thickness (Tb.Th), trabecular separation (Tb.Sp), and trabecular number (Tb.N) (−6.61%, *p* > 0.15; 0.36%, *p* > 0.9; 0.75%, *p* > 0.9; respectively). The septal mineralized cartilage volume (Sept.MCV) and septal mineralized cartilage volume fraction (Sept.MCV/TV) of calcified septa in the notochordal canal of the vertebra showed a large reduction (−73.19%, *p* < 0.003; −77.83%, *p* < 0.001; respectively) in the flight group vs. the control. On the contrary, a statistically significant increase (+35.73 %, *p* = 0.0343) in bone volume (BV) was found for the intercentra after the space flight. Since the intercentra consists almost entirely of cortical bone, the bone volume fraction parameters (BV/TV) were not calculated.

## 4. Discussion

Historically, a variety of organisms have been used as model subjects for space research projects, including rodents, fish, amphibians, and birds [15,26,27]. However, reptiles have been relatively rare objects of study. Geckos, in particular, have been utilized in several orbital experiments due to their high adaptability to space flight conditions and their unique ability to remain attached to surfaces in G0 conditions. During the Foton-M2, Foton-M3, and Bion-M1 missions, the small changes observed in the main organs and skeletal bones of thick-toed geckos were nonpathological and reversible and were likely a result of the feeding strategy, rather than the space flight conditions themselves.

The results of our research on the proximal caudal vertebrae and notochord in TG, which have never been studied before, support previous observations of the gecko skeleton’s resistance to bone loss [8]. In particular, we did not find any meaningful change in the bone volume ratio for cortical or trabecular bone in the proximal caudal vertebrae after the flight. On the other hand, we were the first to note changes in the volume of mineralized notochordal cartilage and intercentra that may be associated with G0. TG showed a statistically significant decrease in the mineralization of notochordal cartilage and an increase in the bone volume of the intercentra after 30 days in space. These facts require further research. While humans lose their notochords as they grow, geckos preserve the notochord throughout their lives, offering a unique opportunity for us to study its structure, development, and relationship with other tissues. We believe that the mineralized tissue of notochord, particularly the notochordal septum, potentially can be a sensible indicator of the mineral balance response to G0 exposure. The volume of calcified tissue in the septum is significantly lower than that of bones, which enables the detection of even slight variations in mineral level. However, the mineralization of notochord-derived cartilage in geckos is a complex process involving the deposition of calcium phosphate crystals within the cartilaginous matrix. Various factors can influence the process, including age, diet, and environmental conditions. For example, a diet high in calcium and phosphorus can promote mineralization, while deficiencies in these nutrients may result in a poorly mineralized notochord cartilage. Additionally, environmental factors, such as temperature and humidity, can also impact the mineralization process [36]. The factors mentioned above, along with the considerable inter-species diversity of geckos, suggest that the new approach proposed by us for notochord mineralization tracking must be further validated through future space experiments that involve a larger number of geckos.

The space flight environment is characterized by the combined effect of G0, air environment, radiation exposure, and overloads during launching and landing. All these stress factors can cause adaptive changes in the metabolism of geckos. The number of publications in this field is limited and further in-depth physiological, histological, and immunohistochemical studies are necessary to better understand how the organism adapts to space flight.

Another concern is the increased bone cell apoptosis in G0. The author of Ref. [34] stated that osteocyte apoptosis in mouse models resulted in decreased osteocyte lacunar volumes and increased lacunar vacancies. According to Ref. [21], the osteocyte lacunae in mouse models displayed a reduction in volume and a shift in shape towards a spherical form during the space flight. Additionally, there was a significant increase in empty lacunae when compared to the control group. Considering the results from the mouse model, further in-depth research on structural changes in gecko bones at the osteocyte level should be carried out. Our research findings indicate that X-ray phase-contrast imaging is an effective method that provides the necessary sensitivity, spatial resolution, and field of view for studying 3D bones and soft tissue morphological structure at the cellular level. This is highlighted in Figure 5, where both chondrocyte cells (solid arrow) in the mineralized notochordal cartilage and osteocyte lacune (dotted arrow) in the vertebral bone are distinctly visible. Additionally, other researchers have demonstrated that synchrotron radiation nano-tomography allows for detailed imaging of the 3D lacunar-canalicular network, as referenced in Ref. [54]. Our future work will involve using X-ray phase-contrast imaging techniques with the TG model to carry out in-depth 3D studies of the G0-induced change in bone ultra-structure.

Other future research efforts should focus on uncovering the molecular mechanisms underlying notochord preservation in geckos. Identifying the genes and signaling pathways involved in this process may provide valuable insights into the evolution of notochord preservation and its role in gecko biology.

The study of gecko bones and their notochord cells in G0 conditions offers valuable insights into the developmental processes and potential health effects related to space flight. The findings from these studies may help improve our understanding of the mechanisms behind bone loss and, potentially, muscle atrophy and other health issues experienced by astronauts in space.

Moreover, the study of geckos and other animals in space research can contribute to our knowledge of the fundamental biological processes that govern adaptation to altered gravity conditions, ultimately leading to the development of novel countermeasures and technologies to support the human exploration of space.

## 5. Conclusions

In mammals, including humans, the lack of support afferentation can lead to demineralization and neuromuscular problems. The results of Foton-M series and Bion-M1 experiments suggest that a gecko’s ability to attach to surfaces and maintain normal locomotion, along with other physiological and metabolic features, contribute to stable mineral metabolism and prevent demineralization in their skeletal bones during prolonged space flight. This makes geckos a valuable animal model for orbital experiments.

Our research focused specifically on the influence of G0 on the most proximal caudal vertebrae and notochord of TG, areas that have not been previously studied. The results of our investigation confirmed previous observations regarding the gecko skeleton’s ability to resist bone loss. Specifically, we observed no statistically significant changes in the bone volume fraction for either cortical (Ct.BV/TV, *p* > 0.2) or trabecular bone (Tb.BV/Sc.V, *p* > 0.7) within the proximal caudal vertebrae following flight. However, we were the first to note a noticeable decrease in mineralization within the notochordal septum (Sept.MCV/TV, −77.83%, *p* < 0.001) and a significant increase in intercentral volume (BV, +35.73%, *p* = 0.0343) after the flight.

The persistent notochord of geckos provides a unique opportunity to study its composition, growth, and connection with other tissues in adult organisms. During development, geckos form cartilage derived from the notochord, which can eventually undergo mineralization. We propose a new approach to monitor the mineral metabolism of TG in G0 by measuring the amount of mineralized tissue in the notochordal septum. This method has the potential to be a sensitive technique to follow the demineralization process in G0, as the volume of septal calcification is limited, allowing even small changes in mineral content to be detected.

## Figures and Tables

**Figure 1 cells-12-02415-f001:**
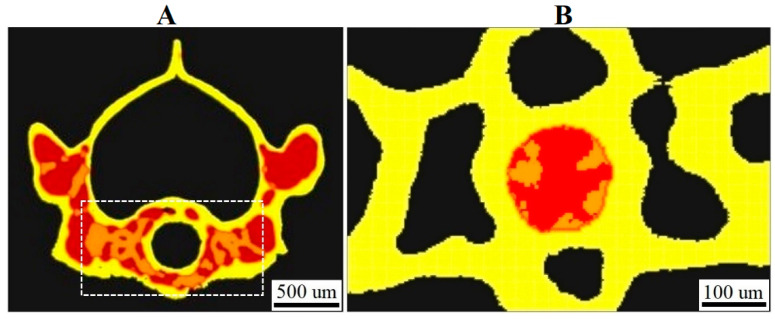
Transverse CT sections of the gecko’s caudal vertebra: (**A**) the segmented cortical bone (Ct.B), yellow; the subcortical volume (Sc.V), red; the ROI in the ventral side of the vertebra segmented for trabecular parameter (Tb.Th, Tb.Sp, Tb.N) calculation, marked by a dotted rectangle; (**B**) the ROI (TV) for segmentation and calculation of the septum mineralized cartilage volume (Sept.MCV) in the notochordal canal of the vertebra, red.

**Figure 2 cells-12-02415-f002:**
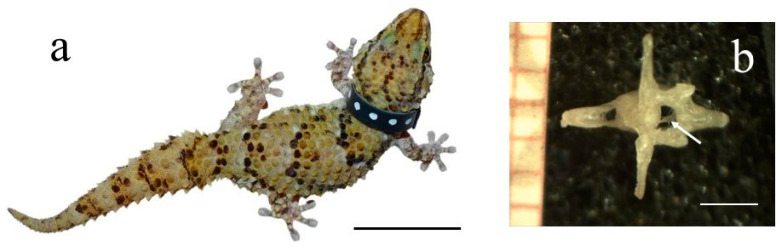
(**a**) A thick-toed gecko that participated in the orbital experiment; (**b**) posterior view of the gecko’s vertebra with the transverse processes, haemal arch, and clearly visible vertebral canal (white arrow). Scale bar (**a**) 3 cm, (**b**) 1.8 mm.

**Figure 3 cells-12-02415-f003:**
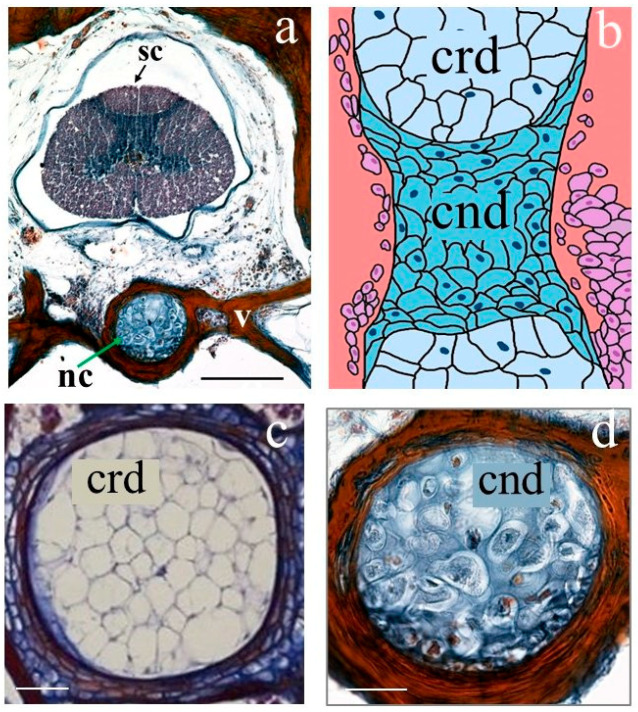
TG proximal caudal fragment’s histological structure: (**a**) transverse section (Mallory staining); (**b**) drawing depicting the longitudinal section of the TG notochord taken from the position of a caudal vertebra; (**c**) transverse sections of the notochord (stained with Mallory’s stain) taken from position indicated in (**b**) with crd; (**d**) transverse section of the notochordal cartilage with cnd (stained with trichrome Mallory’s stain). Nc, notochordal canal; sc, spinal cord; v, vertebra; cnd, chondroid cells/tissue; crd, chordoid cells/tissue. Scale bars (**a**) 500 microns, (**c**,**d**) 50 microns.

**Figure 4 cells-12-02415-f004:**
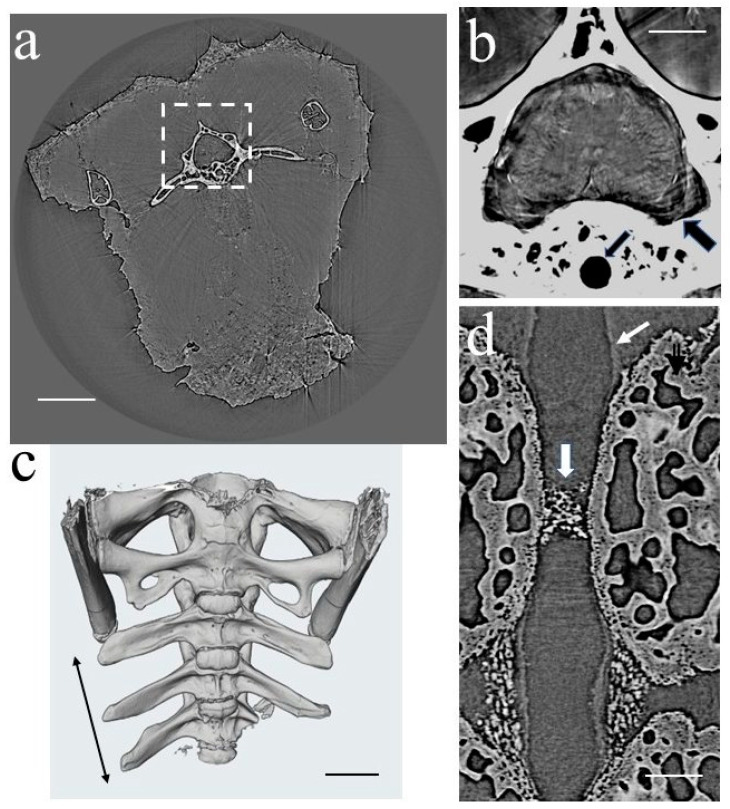
XPCT images of the proximal caudal part of the gecko’s body (pixel size 3.3 μm^2^): (**a**) the transverse section of the specimen, vertebra is boxed with a dashed line; (**b**) gecko’s spinal cord (gray color) surrounded by vertebra (white color), thick and thin black arrows show vertebral and notochordal canals, respectively; (**c**) 3D image of the proximal caudal vertebrae studied in our research (double sided arrow); (**d**) lateral view of the vertebrae with notochordal canal (thin white arrow) and mineralized notochondral cartilage (thick white arrow). Scale bars (**a**) 1.5 mm, (**b**,**d**) 500 micron, (**c**) 1 mm.

**Figure 5 cells-12-02415-f005:**
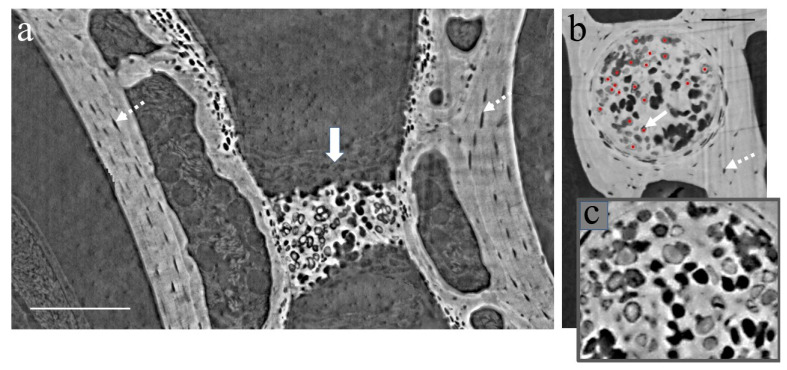
XPCT images (pixel size 0.7 μm^2^): (**a**) longitudinal section of the vertebra with notochordal canals and notochord with mineralized notochordal cartilage (solid white arrow), where the dotted white arrows show osteocyte lacunae in the vertebral bone; (**b**) the transverse section of the vertebrae at mid-vertebral regions corresponding to reduced diameters of the notochord with mineralized notochordal cartilage, where the white arrow indicates chondrocytes, the dotted white arrow shows osteocyte lacunae in the vertebral bone and red dots show chondrocyte nuclei; (**c**) zoomed image of mineralized septum. Scale bars (**a**) 200 microns, (**b**) 100 microns.

**Figure 6 cells-12-02415-f006:**
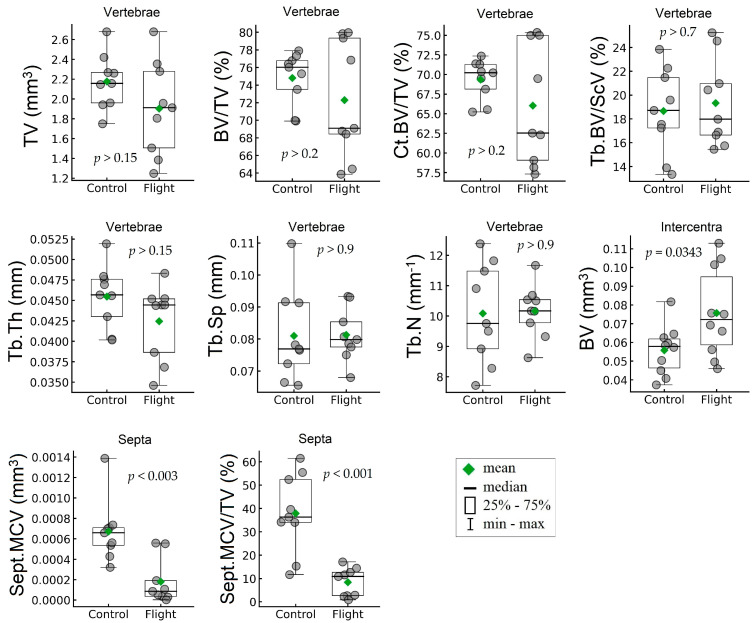
Effects of space flight on morphometric parameters of the geckos vertebrae (C1–C3) and intercentra (IcS2; IcC1–IcC3).

**Table 1 cells-12-02415-t001:** Morphometric parameters of the geckos’ vertebrae in control and flight groups after the experiment.

Index	Units	Control Group		Flight Group		Difference,%	*p*-Value
		Mean	SD	Mean	SD		
Total Body							
Total length, mm	mm	158.0	2.78	150.67	7.52	−4.64	0.2275 (NS)
Snout-vent length, mm	mm	78.33	0.58	79.50	3.50	1.49	0.6579 (NS)
Tail length, mm	mm	79.67	3.33	71.17	6.25	−10.67	0.1277 (NS)
Launch body mass, g	g	20.45	0.80	19.69	2.17	−3.72	0.6156 (NS)
Landing body mass, g	g	19.54	1.25	20.11	1.94	2.90	0.6955 (NS)
Delta body mass at landing, g	g	−0.91	1.11	0.42	1.14	−	0.2232 (NS)
Whole Vertebrae							
TV, mm^3^	mm^3^	2.18	0.28	1.90	0.48	−12.55	0.1608 (NS)
BV/TV, %	%	74.80	3.020	72.30	6.67	−3.35	0.3266 (NS)
Ct.BV/TV, %	%	69.34	2.54	66.033	7.68	−4.77	0.2486 (NS)
Tb.BV/Sc.V, %	%	18.66	3.58	19.33	3.69	3.57	0.7028 (NS)
Ventral side of vertebrae							
Tb.Th, mm	mm	0.046	0.0038	0.043	0.0046	−6.61	0.1505 (NS)
Tb.Sp, mm	mm	0.081	0.014	0.081	0.0082	0.36	0.9580 (NS)
Tb.N, mm^−1^	mm^−1^	10.085	1.64	10.16	0.87	0.75	0.9045 (NS)
Mineralized cartilage of septa							
Sept.MCV, mm^3^	mm^3^	0.00067	0.0003	0.00018	0.00022	−73.19	0.0027
Sept.MCV/TV, %	%	37.86	16.98	8.40	6.17	−77.83	0.0006
Intercentra							
BV, mm^3^	mm^3^	0.056	0.013	0.076	0.024	35.73	0.0343

NS, not significant; SD, standard deviation.

## Data Availability

All data generated or analyzed during this study are included in this article.

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
