# Peer review of "The Study of the Caudal Vertebrae of Thick-Toed Geckos after a Prolonged Space Flight by X-ray Phase-Contrast Micro-CT"

_cells, 2023, doi:10.3390/cells12192415_

Round 1
Reviewer 1 Report
The review ‘The study of the caudal vertebrae of thick-toed geckos after a prolonged space flight by X-ray phase-contrast micro-СТ’ written by Bukreeva et al. is interesting for scientists working in the field of space research and for space interested people.
The authors studied the proximal caudal vertebrae in thick-toed geckos after a 30-day space flight onboard the biosatellite Bion-M1. They used a synchrotron-based X-ray phase-contrast micro-CT 22 (XPCT) for their experiment. Interestingly, they found no statistically significant changes in the main bone parameters, while a significant decrease in the mineralization of notochordal cartilage and an increase in the bone volume of the intercentra were found.
This is a well-written, very interesting manuscript. The authors present interesting data in the field of space research. I like to congratulate the authors for this nice space experiment.
I have no concerns. This manuscript is acceptable for publication in Cells.
1. What is the main question addressed by the research? A: Study of the notochordal septa calcification to observe the impact of long-term microgravity on mineral metabolism, could prove to be beneficial in space exploration related to bone loss. 2. Do you consider the topic original or relevant in the field? Does it address a specific gap in the field? A: Yes, the topic is original and relevant for the field of space research 3. What does it add to the subject area compared with other published material? A: Studying of geckos and other animals in space research can increase the current knowledge of biological processes changed in microgravity. This can contribute to develop new countermeasures and technologies to support space exploration. 4. What specific improvements should the authors consider regarding the methodology? What further controls should be considered? A: The methodology is appropriate for this study. They used a synchrotron-based X-ray phase-contrast micro-CT 22 (XPCT) for their experiment. 5. Are the conclusions consistent with the evidence and arguments presented and do they address the main question posed? A: yes, they are. 6. Are the references appropriate? A: yes 7. Please include any additional comments on the tables and figures. A: Tables and figures are fine.
Author Response
Dear Reviewer.
Our responses to the review are in the attached file
With respect,
Yuri Krivonosov

Reviewer 2 Report
The Article « The study of the caudal vertebrae of thick-toed geckos after a prolonged space flight by X-ray phase-contrast micro-СТ« Bukreeva I. et al. is very interesting.
The authors aimed to study of the caudal vertebrae of thick-toed geckos after a prolonged space flight by X-ray phase-contrast micro-СТ.
The manuscript is not quite an original research article...
The most data were previously published by the same authors (Int. J. Mol. Sci. 2019, 20(12), 3019; https://doi.org/10.3390/ijms20123019)
In additional, in the discussion, the authors’ hypothesized conclusions are repeated.....
"…suggest that gecko’s ability to attach to surfaces and maintain normal locomotion, along with other physiological and metabolic features, contribute to stable mineral metabolism and prevent demineralization in their skeletal bones during prolonged weightlessness..."
Some extensive editing is necessary regarding the introduction and the conclusion.
If statistically derived data is available, this should be accented.
Author Response

(The authors gave the same response as above.)
